# Lifestyle Modification: Evaluation of the Effects of Physical Activity and Low-Glycemic-Index Mediterranean Diet on Fibrosis Score

**DOI:** 10.3390/nu15163520

**Published:** 2023-08-10

**Authors:** Ritanna Curci, Antonella Bianco, Isabella Franco, Caterina Bonfiglio, Angelo Campanella, Antonella Mirizzi, Vito Giannuzzi, Raffaele Cozzolongo, Nicola Veronese, Alberto Ruben Osella

**Affiliations:** 1Laboratory of Epidemiology and Statistics, National Institute of Gastroenterology—IRCCS “S. de Bellis”, Via Turi, 70013 Castellana Grotte, Italy; ritanna.curci@irccsdebellis.it (R.C.); antonella.bianco@irccsdebellis.it (A.B.); isabella.franco@irccsdebellis.it (I.F.); catia.bonfiglio@irccsdebellis.it (C.B.); angelo.campanella@irccsdebellis.it (A.C.); 2Via Fratelli Morea, 23, 70017 Putignano, Italy; dottoressamirizzi@yahoo.com; 3Gastroenterology Unit, National Institute of Gastroenterology—IRCCS “S. de Bellis”, Via Turi, 70013 Castellana Grotte, Italy; vito.giannuzzi@irccsdebellis.it (V.G.); raffaele.cozzolongo@irccsedebellis.it (R.C.); 4Geriatric Unit, Department of Medicine, University of Palermo, 90100 Palermo, Italy; nicola.veronese@unipa.it

**Keywords:** liver fibrosis, non-alcoholic fatty liver disease, physical activity, Mediterranean diet, lifestyle, chronic liver disease

## Abstract

Background: Non-Alcoholic Fatty Liver Disease (NAFLD) is one the most prevalent causes of chronic liver disease worldwide. In the absence of an approved drug treatment, lifestyle modification is the first intervention strategy. This study aimed to estimate the main effect of two different physical activity (PA) programs, and a Low-Glycemic-Index Mediterranean Diet (LGIMD), or their combined effect on liver fibrosis parameters in subjects with NAFLD. Methods: Subjects with moderate or severe NAFLD grade of severity (n = 144) were randomly assigned to six intervention arms for three months: LGIMD, PA programs, and their combination. Data were collected at baseline, 45 days, and 90 days. Transient elastography was performed to assess the outcome. Results: at 90 days, a statistically significant reduction in kPa was found among subjects following LGMID (−2.85, 95% CI −5.24, −0.45) and those following an LGIMD plus PA1 (−2.37, 95% CI −4.39, −0.35) and LGIMD plus Pa2 (−2.21, 95% CI −4.10, −0.32). The contrast between time 2 and time 1 of the LGIMD plus PA2 treatment showed a statistically significant increase, and vice versa: the contrast between time 3 and time 2 of the same treatment showed a statistically significant reduction. The PA1 and PA2 arms also showed reduced kPa, although the results did not reach statistical significance. Conclusions: The intervention arms, LGIMD, LGIMD+PA1, and LGIMD+PA2, reduced the fibrosis score.

## 1. Introduction

Non-Alcoholic Fatty Liver Disease (NAFLD) is the most common liver disease, affecting about one third of the world’s population [1]. It is considered a phenotype of a metabolic syndrome that occurs in the liver [2]. NAFLD encloses a broad spectrum of diseases including non-alcoholic fatty liver (NAFL) as well as non-alcoholic steatohepatitis (NASH), in which inflammation and fibrosis develop [3]. NASH can eventually progress to cirrhosis and hepatocellular carcinoma. Therefore, the fibrosis severity becomes a key prognostic factor [4]. The presence of fibrosis is the most relevant feature associated with liver-related outcomes such as mortality [5]. Two key components of chronic liver disease, portal hypertension and failure of hepatocyte function, are promoted by fibrosis. Several other adverse clinical factors including cardiovascular events, ischemic stroke, metabolic complications, and cardiovascular mortality are associated with the presence of liver fibrosis [6]. As a consequence, early recognition and management of liver fibrosis is critical to slow or prevent progression. Lifestyle modifications, mainly diet and PA, are the first line of intervention for NAFLD/NASH as the pharmacological management is very poor [7]. A weight loss of 10% or more has shown an improvement of fibrosis when a PA program is included [8]. Recently, the association between a Mediterranean diet (MD) and the risk of liver cirrhosis has been identified by a population-based study in adults with NAFLD [9]; the results evidenced significatively better metabolic control with a high adherence to MD. In obese subjects, an improvement of fibrosis has been shown with weight loss [8]; weight is considered an independent predictor for progression of steatosis to liver fibrosis in NAFLD [10]. The beneficial effects of MD on patients with NAFLD are well known; however, the relationship between MD and liver fibrosis remains to be investigated. Although the impact of PA on liver fibrosis is not yet clear and an optimal protocol in terms of the PA parameters’ frequency, intensity, type, and time (FITT) has not yet been defined, it is known to influence the development and progression of fibrosis in NAFLD subjects [3]. Previous studies have shown a reduction in fibrosis in obese subjects with NAFLD who followed an aerobic high-intensity interval training program, resulting in an improvement in liver stiffness of −16.8% [11,12]. A cross-sectional study conducted by the NASH Clinical Research Network showed that subjects who followed recommendations for vigorous physical activity (6 MET h/week, lasting at least 75–150 min/week) exerted greater health benefits as compared with subjects who followed recommendations for moderate activity (3–5.9 MET h/week, during at least 150–300 min/week), including a significant reduction in advanced fibrosis and a lower likelihood of developing NASH [13].

Since 2010, the Laboratory of Epidemiology and Statistics at IRCCS “S. de Bellis” has been studying the effects of lifestyle changes on NAFLD. We have studied several aspects of these changes including the effects of an LGIMD alone [14], aerobic versus resistance PA [15], and the combination of diet and PA [16]. Specifically, we showed that the combination of LGIMD and aerobic PA is the most effective treatment for individuals with NAFLD [16]. In addition, we highlighted the effect of combined PA and LGIMD on glucose metabolism parameters at different grades of NAFLD severity [17].

In this context, we hypothesized that an aerobic or resistance PA program and LGIMD could have a positive effect on liver fibrosis parameters in subjects with NAFLD. In this study, as secondary analysis, we aimed to estimate the effects of a 3-month program of aerobic and resistance PA alone or combined with LGIMD, compared with LGIMD alone on the kPa, as measured by transient elastography.

## 2. Materials and Methods

### 2.1. Participants

Details about the primary study have already been published elsewhere [16]. Briefly, the NUTRIATT (NUTRItion and Ac-TiviTy) study aimed to estimate the effect of diet, PA, and their combination on NAFLD in subjects aged 30–60 years old. To do this, we enrolled patients referred to our IRCCS by the local General Practitioners. It was conducted from March 2015 to December 2016 at the National Institute of Gastroenterology, IRCCS “S. de Bellis” Research Hospital, Castellana Grotte (BA), Italy. Once enrolled and randomized, the participants were followed up by trained Nutritionist and Motor Science graduates throughout the complete follow-up. The trial, which was registered at clinicaltrials.gov (CT0234747696), was conducted in accordance with the Declaration of Helsinki and was approved by the Ethics Committee (Prot. No. 10/CE/De Bellis, 3 February 2015).

### 2.2. Study Design

This was a parallel group randomized controlled clinical trial. The inclusion criteria were (1) Body Mass Index (BMI) ≥ 25.0, (2) participants aged 30–60 years old, and (3) presence of moderate or severe NAFLD as assessed by FibroScan^®^. The exclusion criteria included (1) overt cardiovascular disease, (2) stroke, (3) clinically evident peripheral arteriopathy, (4) diabetes mellitus type II (oral antidiabetic therapy, blood glucose > 126 mg/dL, random blood glucose finding > 200 mg/dL), (5) medical conditions that could reduce the subject’s ability to follow a diet, (6) impossibility of following an LGIMD for various reasons such as religion, and (7) contraindications to the practice of PA.

### 2.3. Sample Size

As this is a secondary analysis, the sample size was not estimated for the main outcome of the registered trial. However, we calculated that for a reduction from 8 to 7 points, as measured by the kPa parameter, shifting the Type II probabilistic error from 80 to 95% and setting the Type I probabilistic error to 0.05, 0.01, and 0.001, the sample size range would equal n1 = n2 = n3 = n4 = n5 = n6 = 20–30; with a power of 97% and Type I probabilistic error of 0.01 the sample size was estimated as 20 participants per group.

### 2.4. Data Collection

During enrollment, data about the socio-demographic, lifestyle, and medical history aspects of the subjects were collected. The International Physical Activity Questionnaire Long Form (IPAQ-LF) was used to collect information on PA [18] and information about alcohol intake and eating behavior was collected by using the European Prospective Investigation into Cancer and Nutrition Food Frequency Questionnaire (EPIC FFQ) [19]. Anthropometric measurements (weight, height, waist circumference) were taken by trained dietitians in a standard manner [20]. The subjects wore only underwear during the measurements. SECA instruments (Model 700 and Model 206; 220 cm; SECA, Hamburg, Germany) were used to take weight and height measurements, respectively.

Bioelectrical Impedance Analysis (BIA) was used to indirectly determine body composition by measuring the impedance, given by resistance (R) and reactance (Xc). Bioelectrical impedance was measured with a phase-sensitive touch-screen impedance device (Nutrilab™, Akern, Florence, Italy), working with alternating sinusoidal electric current of 245 microampere at an operating frequency of 50 kHz (±1%).

All subjects were asked to remain in a supine position with a leg opening of 45° compared with the median line of the body and the upper limbs positioned 30° away from the trunk. After cleansing the skin, two electrodes were placed on the back of the right hand and two electrodes on the corresponding foot, with a distance of 5 cm between them. The electrodes were Ag/AgCl low-impedance electrodes (Biatrodes, Akern Srl, Florence, Italy). As fluid distribution disturbances may occur, subjects were advised to abstain from food and drink for >2 h before the test. After overnight fasting, blood samples were drawn in tubes containing anticoagulant ethylenediamine tetra acetic acid (K-EDTA). Measurements were taken three times during the study (at baseline, after 45 days, and after 90 days) and performed using standard methods.

### 2.5. Outcome Assessment

The controlled attenuation parameter score (CAP) was applied to measure and quantify hepatic steatosis. CAP measures the rate of ultrasound attenuation (due to liver fat) at the standardized frequency of 3.5 MHz through vibration-controlled elastography (VCTE). Following the literature, cut-off points were set at <248, 248–267, 268–279, and ≥280 dB/m for absent, mild, moderate, or severe NAFLD, respectively [21]. Liver fibrosis was assessed by FibroScan^®^ (Echosens, Paris, France) and expressed by the Liver Stiffness Measurement (LSM) Youden cutoff value in kPa [22]; values of >8, 8–12, >12 kPa indicate a low, indeterminate, and high risk of fibrosis, respectively. All quality requirements for Fibroscan^®^ measurements were complied with [23].

### 2.6. Randomization and Masking

Patients were randomized to 1 of 6 treatment groups according to random number sequences given by the Stata user-written program *-ralloc-*. The 6 working arms were: (1) Control Diet (CD) based on the guidelines of the Research Center for Food and Nutrition, Council for Agricultural Research and Economics, Rome, Italy (CREA-AN) (25), (2) LGIMD, (3) aerobic PA program (PA1), (4) combined PA program (aerobic and resistance training) (PA2), (5) LGIMD+PA1 and (6) LGIMD+PA2.

Blinding was maintained in several ways. First, we assured the staff and participants that the interventions (diet and physical activity program) were based on healthy principles. Moreover, during follow-up at each visit the dietitian was assigned on a random daily basis. Furthermore, the different intervention arms were never mixed as only one was called in each day and only one patient at each date, to reduce the information exchange among participants. Physical activity training sessions were also held for only one intervention arm at a time.

Staff members who assessed outcomes were unaware of the intervention assigned to each participant. Two radiologists performed the outcome measurements but only one performed the assessment each day and this order was also randomly assigned. During the outcome measurements, performed at the 45th and 90th day, the radiologists were unaware of the previous measurements.

### 2.7. Dietary Interventions

Two different diets were assigned: CD based on the CREA-AN guidelines [24] and LGIMD [14]. No indication of total caloric intake was imposed. The proposed diets were included in a brochure indicating foods to be consumed always, sometimes, and never. The brochure is given in Appendix A. All participants had to record what they ate in a daily diary. To assess adherence to the LGIMD and the CD diets, the Mediterranean Adequacy Index (MAI) was chosen as the relevant measure [25]. Details on the LGIMD and CD are included in Appendix A.

### 2.8. Physical Activity Interventions

To determine physical fitness and create the right training program, all subjects took field motor tests based on cardiorespiratory fitness [26], strength [27], and flexibility [28]. Subjects randomized to the physical activity intervention arms repeated the tests at the baseline and the two scheduled follow-up dates, whereas subjects included in the diet-only intervention arms took the tests at baseline and after 90 days. Two different PA interventions were featured: PA1, consisting of an aerobic program and PA2, consisting of a combination of aerobic- and resistance-type activity. The maximum age-predicted heart rate was determined by using Tanaka’s formula [29]. Intensity progressions in the exercise programs were decided based on the test results obtained.

#### 2.8.1. Aerobic Activity Program

There were three non-consecutive weekly aerobic training sessions of moderate intensity (60–75% maximum HR, 3.0–5.9 with the following MET distribution: week 1–4: 14.2 kcal ∗ kg^−1^ ∗ week^−1^; week 5–8: 18.9 kcal ∗ kg^−1^ ∗ week^−1^; week 9–12: 23.6 kcal ∗ kg^−1^ ∗ week^−1^). Participants mainly practiced treadmill walking, cycling, cross-training, and rowing. An exercise session lasted between 50 and 60 min. An automatic heart rate monitor was used to monitor the aerobic exercise intensity every 5 min. A weekly total of 150/180 was the duration of the training.

#### 2.8.2. Combined Aerobic and Resistance Activity Program

There were three non-consecutive sessions consisting of (1): 45 min of moderate intensity aerobic exercise with the same characteristics as 2.8.1 and (2) 3 sets of 12 exercises, each until fatigued, on different machines, namely leg press, adductor/abductor machine, gluteal machine, biceps curl, triceps extension, three different abdominal exercises, leg machine, low rower, shoulder flexion. An increase in weight of 1–2.5 kg ∗ week^−1^ when subjects succeeded in completing 10 repetitions in excellent form was carried out. The total duration of weekly exercise was 180/240 min. Both types of PA were carried out at a local gym, and the planning of exercise sessions was decided by the specialist together with the subjects participating.

### 2.9. Statistical Analysis

Description and testing of intervention arms and socio-demographic, lifestyle, and biological variables are reported as means (SD) or frequencies (%), analyzed by t-test or χ2 test, as appropriate. For statistical purposes and following literature recommendations, the measurement of kPa was categorized as Low Risk (<8 kPa), Indeterminate Risk (8–12 kPa), or High Risk (>12 kPa) of liver stiffness. A Generalized Estimating Equation (GEE) [30] was performed to estimate the average change in kPa over time while controlling for several covariates. GEE models are particularly useful in health studies to estimate mean changes in outcome measurements while controlling for covariates. The procedure makes it possible to probe for correlations of response data (repeated measurements on each subject). By using diagnostic tools, the best-performing set of covariates and working correlation structures (SE/robust options) was chosen. Model selection was made using diagnostic tools. The outcome variable was not normally distributed; thus, a gamma distribution with identity link for the outcome was assumed and an unstructured correlation matrix was set to the data. Gender, Age, Systolic Blood Pressure (SBP), Waist-to-Hip Ratio (WHR), Body Mass Index (BMI), Arachidonic Acid Eicosapentaenoic acid ratio (AA_EPA), Glutamic-Oxalacetic Transaminase/Glutamic-Pyruvic Transaminase ratio (GOT/GPT) were taken considered as adjusting covariates. We choose to perform the statistical analysis on a intention-to-treat basis. All the results are expressed in the original metric as mean ± 95% Confidence Interval (95% CI). The outcome predictions obtained using post-estimation tools were subsequently graphically displayed. Stata statistical software v. 18.0 (Stata Corp, 4905 Lakeway Drive, College Station, TX, USA) was used to perform statistical analyses. The official command *-xtgee-* and the user-written contribution *-QIC-* (criterion for model selection in GEE analyses) were used.

## 3. Results

The flowchart of the study is shown in Figure 1. All drop-outs were due to desertion. The study subjects’ (n = 144) socio-demographic characteristics are shown in Table 1. Mean age was equally distributed in the groups (50 ± 55) except in the PA2 and PA2 plus LGIMD arms, where it was slightly lower (46 ± 9 and 46 ± 10, respectively). Most of the participants were men (62%) and married (67%). The predominant level of schooling in the sample was high school (49%) and most of the subjects had never smoked (70%).

Anthropometric, body composition, and biochemical characteristics of participants are shown in Table 2.

Overall, the cut-off value of the kPa parameter at 90 days was lower than at baseline in the LGIMD, PA1 plus LGIMD, and PA2 plus LGIMD groups, whereas in the other groups an increase from the baseline score was observed. The CAP decreased at 90 days in all groups compared with the baseline value. There were no differences between the characteristics of the participants in the six intervention arms except for the mean BMI assessed at baseline, which was higher in the CD arm and the PA2 plus LGIMD arm (34 ± 4 and 34 ± 3 kg/m^2^, respectively). Glucose values had decreased in all treatment groups by the end of treatment compared with baseline except in the LGIMD group; the same applied to HOMA-IR in the treatment groups at 90 days, except in the PA1 plus LGIMD.

GEE analysis results are shown in Table 3. There was a statistically significant effect of Time at the 90th day (1.79, 95% CI 0.21, 3.38) and LGMID (2.94, 95% CI 0.67, 5.22) on kPa. Estimated modification effects between time and diets showed a statistically significant reduction in kPa among subjects who followed the LGMID (−2.85, 95% CI −5.24, −0.45) and those who followed LGIMD plus PA1 (−2.37, 95% CI −4.39, −0.35) and LGIMD plus PA2 (−2.21, 95% CI −4.10, −0.32) at the 90th day. In the PA1 and PA2 arms, kPa also decreased, although this did not reach statistical significance.

The results of GEE analysis are graphically displayed in Figure 2.

Post-estimation analyses are shown in Table 4. The contrasts between the mean expected values for the effects of each intervention at different times in kPa are demonstrated. The contrast between time 2 and time 1 of the LGIMD plus PA2 treatment showed a statistically significant increase, and vice versa: the contrast between time 3 and time 2 of the same treatment showed a statistically significant reduction (1.264, 95% CI 0.034, 2.494 and −1.682, 95% CI −2.873, −0.490, respectively).

## 4. Discussion

Being overweight or obese promotes the development of NAFLD, a progressive liver disease that could include simple steatosis, NASH, fibrosis, and cirrhosis without excessive alcohol consumption (<20 g day^−1^) [31].

Recently, fibrosis of adipose tissue has been shown to occur in obesity, along with inflammation [32] and these facts have been shown to be causally linked to the pathogenesis of several chronic inflammatory diseases, such as atherosclerosis and NASH [33]. Our results showed that time and diet alone or in combination with PA induced a modification effect that was statistically significant in reducing kPa at the 90th day (although there was an increase at the 45th day) among subjects who followed LGMID and those who followed LGIMD plus PA1 and LGIMD plus PA2.

Many studies have shown the crucial role of lifestyle changes in the management of NAFLD, in particular on the degree of liver fibrosis [34]. Dietary habits and PA contribute to reducing fibrosis and improving individuals’ quality of life [3]. The NAFLD practice guidelines of the American Association for the Study of Liver Diseases (AASLD) advise that a body-weight loss of at least 3% to 5% decreases hepatic steatosis, and histologic improvements (including fibrosis) are seen with a greater body-weight reduction (7%) [35].

In the LGIMD arm, kPa tended to decrease gradually up to 90 days without ever increasing from baseline. Myrian et al. confirmed in their study that adhering to a Mediterranean diet program reduces the risk of fibrosis in subjects with NAFLD [9]. MD helps to improve metabolic health and weight control thanks to an increased intake of whole grains, fruits, vegetables, legumes, nuts, and fish and a reduced intake of red and processed meats and dairy products [9]. The high fiber content of MD has been shown to protect against becoming overweight or obese by reducing feelings of hunger, as well as through calorie density due to its low glycemic index [36]. Furthermore, the supply of antioxidants such as vitamins, selenium, and polyphenols contributes to a reduced production of proinflammatory mediators and ROS [37].

MD limits the consumption of processed [38] and high fructose foods [39] and limits the excessive intake of advanced glycation end products (AGEs) (a heterogeneous class of non-enzymatic products derived from protein, lipid, and nucleic acid glycation) [40]. AGEs are involved in the etiology pathway of diabetes and other metabolic disorders [38]. It has been shown that AGEs are increased in NASH patients, and they are correlated positively with insulin resistance and negatively with adiponectin [41].

The LGIMD satisfies these requirements and it has shown to be effective against fibrosis at 90 days compared with the control diet arm.

Nevertheless, the relationship between the MD and fibrosis is still to be clarified and there is no evidence that diet itself could improve liver fibrosis; moreover, it is necessary to adopt a sustainable dietary program in the long term to obtain excellent benefits [7].

In our study, both aerobic and combined PA programs improved kPa at 90 days although this effect did not reach statistical significance. However, these results are supported by some studies which evidenced that PA alone in the management of NAFLD is not sufficient, as it produces only a minimal reduction in body weight [12,42], although it improves the liver dysfunction underlying NAFLD including fat accumulation, inflammation, and fibrosis [43].

Junichi et al. reported that using PA as therapy slowed the progression of fibrosis in NAFLD patients by improving glucose metabolism, reducing oxidative stress, and keeping skeletal muscle mass functioning [44]. Therefore, this suggests that low levels of PA increase the risk of liver fibrosis in individuals with NAFLD [3], but further research is needed to disentangle the effect of PA alone on fibrosis.

A few studies have probed the effects of the combination of a diet and PA on fibrosis, but there is little evidence about the beneficial effects of PA and diet intervention on NAFLD, in terms of a suppression of fatty liver disease and its associated mechanisms [42]. Lifestyle modification through a combination of diet and PA produces a significant reduction in hepatic glyceride content (HTGC) and inflammation with a 10% weight loss in subjects with NAFLD [8]. It has been proposed that the mechanism involved in the change in HTGCs after PA could be due to changes in energy balance, circulating lipids, and insulin sensitivity [45].

Visceral fat is directly related to liver inflammatory status and fibrosis, independent of insulin resistance and hepatic steatosis [46]. Van der Poorten et al. showed that an influx of fatty acids and synthesis of cytokines and adipokines promotes hepatic lipid accumulation, insulin resistance, and inflammation [46]. Other studies showed that CK-18, a marker of apoptosis, is related to liver damage and fibrosis [47]; PA promoted the reduction in CK-18 levels and other circulating cytokines in subjects with NAFLD, thus playing a protective role [48]. This was also evident in our study, as we found that after 90 days, there was a statistically significant decrease in kPa in both the intervention arms: LGIMD plus aerobic PA and LGIMD plus combined PA.

Aerobic PA improves NAFLD by activating lipolysis in various tissues and altering adipokine levels [49], and improves markers of hepatic injury such as aspartate amino transferase and alanine-aminotransferase levels, and histologic features of NAFLD [45]. Indeed, we verified in our study that aerobic PA and LGIMD used as treatments promoted an improvement in NAFLD and a decrease in the fibrosis value in the treated subjects. In the combined PA with LGIMD intervention arm, fibrosis markers increased significantly up to 45 days and then decreased by the end of treatment. Resistance activity improves NAFLD by promoting hypertrophy of type II muscle fibers, alters myokine levels, and activates glucose transporter 4 [49]. Hallswort et al. reported that resistance PA improves NAFLD regardless of body weight changes; 8 weeks of PA reduced intrahepatic triglyceride levels by 13% [50].

Overall, a valid approach indicates that the beneficial effects of PA on NAFLD and associated conditions are greater when exercise is combined with a dietary regimen [42,51].

Several methodologic issues need to be considered. In this study, interventions were allotted under supervision by trained personnel both in nutrition and physical activity. Furthermore, PA programs were built following specific FITT parameters as recommended, and adherence to the nutritional program was evaluated with a valid instrument. Limitations include the small number of patients treated and the short intervention time, especially to revealing evident changes in the fibrotic markers. However, the estimates obtained are reliable because we implemented an intention-to-treat strategy.

## 5. Conclusions

Lifestyle modifications are the main interventions recommended in NAFLD patients. In this study, we show that LGIMD, LGIMD plus aerobic PA performed at an intensity of 60–75% of HRmax, and LGIMD plus combined PA performed for a total time of at least 180 min per week for 3 months both reduced the degree of fibrosis in individuals with NAFLD. These activities should be included in primary prevention programs for NAFLD and associated pathological conditions. It is important to be able to rely on a multidisciplinary team to manage subjects with NAFLD and to encourage them to make such lifestyle changes.

## Figures and Tables

**Figure 1 nutrients-15-03520-f001:**
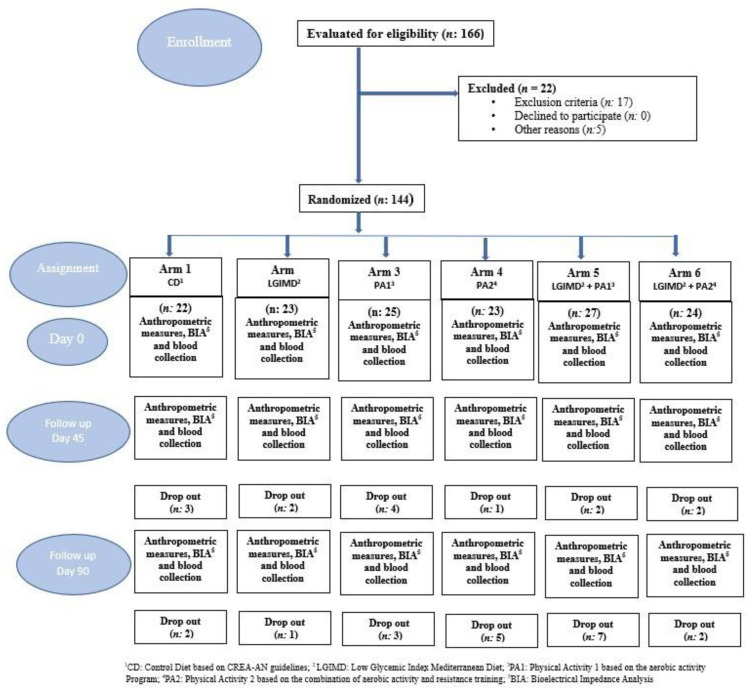
Flowchart of study design.

**Figure 2 nutrients-15-03520-f002:**
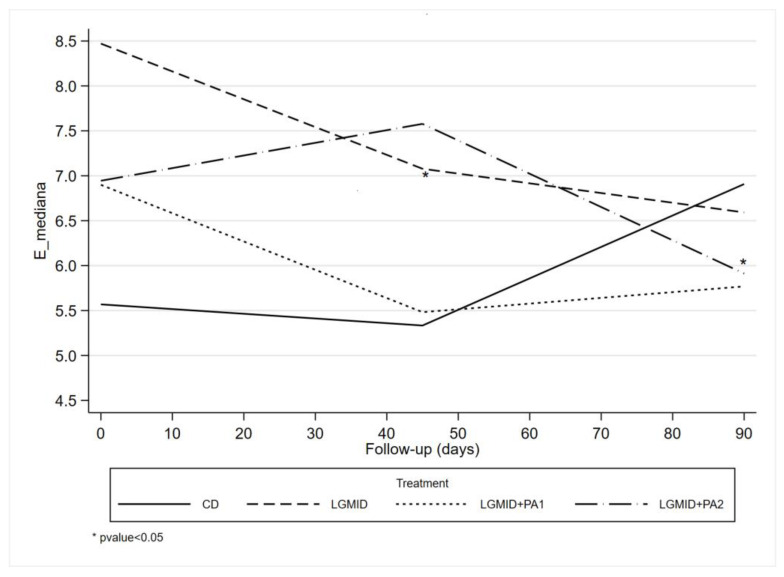
GEE: Marginal Means by Follow-up Time and Intervention Arm. NUTRIATT Study.

**Table 1 nutrients-15-03520-t001:** Socio-demographic characteristics of Participants by Intervention Arm, NUTRIATT Trial, Castellana Grotte, 2015–2016.

	Working Arms					
Variables	C ^a^	LGIMD ^b^	PA1 ^c^	PA2 ^d^	PA1+LGIMD ^e^	PA2+LGIMD ^f^
N	22	23	25	23	27	24
Gender						
Male	11 (50%)	13 (57%)	14 (56%)	17 (74%)	18 (67%)	16 (67%)
Female	11 (50%)	10 (43%)	11 (44%)	6 (26%)	9 (33%)	8 (33%)
Age (Years)	50.70 (8.67)	50.74 (1.75)	50.45 (9.45)	46.23 (9.39)	50.32 (9.61)	36.75 (10.50)
Single	1 (6%)	1 (6%)	1 (5%)	3 (19%)	2 (10%)	3 (16%)
Married	17 (94%)	17 (94%)	18 (90%)	11 (69%)	17 (81%)	16 (84%)
Divorced	0 (0%)	0 (0%)	1 (5%)	0 (0%)	2 (10%)	0 (0%)
Widowed	0 (0%)	0 (0%)	0 (0%)	2 (13%)	0 (0%)	0 (0%)
Study level						
Elementary	1 (6%)	2 (11%)	0 (0%)	0 (0%)	1 (5%)	0 (0%)
Secondary school	10 (56%)	8 (42%)	8 (40%)	3 (18%)	4 (19%)	6 (30%)
High school	6 (33%)	7 (37%)	9 (45%)	9 (53%)	8 (38%)	12 (60%)
Diploma university	1 (6%)	1 (5%)	0 (0%)	0 (0%)	0 (0%)	0 (0%)
University degree	0 (0%)	1 (5%)	3 (15%)	5 (29%)	8 (38%)	2 (10%)
Smoking status						
Never smoked	13 (68%)	10 (50%)	12 (55%)	10 (56%)	17 (71%)	12 (55%)
Former smoker	5 (26%)	6 (30%)	5 (23%)	4 (22%)	4 (17%)	4 (18%)
Current smoker	1 (5%)	4 (20%)	5 (23%)	4 (22%)	3 (13%)	6 (27%)

^a^ C: Control Diet based on CREA-AN guidelines; ^b^ LGIMD: Low-Glycemic-Index Mediterranean Diet; ^c^ PA1: Physical Activity 1 based on the Aerobic Activity Program; ^d^ PA2: Physical Activity 2 based on the combination of Aerobic Activity Program and Resistance Training; ^e^ PA1+LGIMD; ^f^ PA2+LGIMD.

**Table 2 nutrients-15-03520-t002:** Anthropometric, Body Composition, and Biochemical Characteristics of Participants on Intervention Arm and Follow-up Time. NUTRIATT Trial, Castellana Grotte, 2015–2016.

Variables	Follow-Up	Working Arms						
		C ^a^	LGIMD ^b^	PA1 ^c^	PA2 ^d^	PA1+LGIMD5 ^e^	PA2+LGIMD ^f^	*p*-Value
N	Baseline	22	23	25	23	27	24	
E (kPa) ^g^	Baseline	5.95 (2.12)	7.35 (4.43)	6.48 (2.66)	6.17 (2.24)	6.48 (3.48)	6.55 (3.76)	0.79
	45 days	5.29 (1.45)	6.74 (2.94)	6.55 (2.99)	5.46 (1.65)	5.25 (2.07)	7.23 (6.82)	0.31
	90 days	7.04 (4.62)	6.17 (1.71)	6.88 (2.71)	6.39 (2.38)	5.89 (3.17)	5.47 (2.25)	0.57
CAP (dB/m) ^h^	Baseline	349.09 (60.28)	341.57 (45.17)	339.92 (44.89)	334.78 (51.01)	328.56 (52.05)	323.63 (53.98)	0.58
	45 days	322.12 (83.18)	296.52 (51.10)	295.10 (50.68)	299.55 (63.73)	275.61 (75.84)	289.38 (70.15)	0.39
	90 days	307.80 (73.10)	303.72 (81.70)	287.88 (54.59)	284.11 (62.22)	259.84 (57.01)	289.18 (72.96)	0.21
SBP (Hg mm) ^i^	Baseline	126.14 (12.14)	128.86 (17.86)	123.80 (13.17)	118.00 (13.22)	127.04 (13.82)	122.71 (17.00)	0.20
	45 days	121.67 (16.80)	120.48 (14.31)	119.50 (10.50)	116.25 (12.66)	120.87 (13.03)	117.08 (13.34)	0.75
	90 days	122.33 (12.37)	122.78 (12.27)	119.06 (10.04)	116.25 (10.88)	123.61 (12.34)	118.41 (12.95)	0.40
DBP (Hg mm) ^l^	Baseline	81.59 (8.51)	82.73 (9.73)	80.60 (6.18)	79.50 (6.26)	82.59 (6.10)	81.04 (9.32)	0.73
	45 days	80.28 (7.37)	79.76 (8.14)	79.50 (8.57)	79.00 (7.88)	80.00 (7.39)	77.71 (5.51)	0.89
	90 days	81.67 (8.16)	77.50 (9.74)	80.63 (6.80)	79.69 (6.94)	81.94 (6.89)	77.50 (6.50)	0.29
BMI ^m^	Baseline	34.14 (4.97)	32.91 (4.26)	32.73 (5.37)	30.79 (3.16)	33.07 (4.20)	34.09 (3.77)	0.13
	45 days	32.50 (4.57)	30.70 (4.22)	31.87 (4.96)	30.05 (3.27)	31.50 (3.61)	32.19 (3.42)	0.35
	90 days	32.12 (5.11)	30.14 (4.73)	31.01 (4.69)	29.80 (3.53)	30.89 (3.91)	31.97 (3.67)	0.53
WC (cm) ^n^	Baseline	106.59 (11.80)	102.91 (10.98)	103.84 (8.58)	99.70 (8.10)	103.48 (11.89)	104.46 (12.15)	0.42
	45 days	102.11 (12.24)	97.24 (10.19)	101.43 (9.71)	97.32 (6.76)	97.92 (9.06)	99.96 (10.20)	0.42
	90 days	101.00 (11.47)	95.72 (9.69)	99.41 (8.71)	97.16 (6.54)	98.01 (8.91)	98.91 (10.57)	0.66
HC (cm) ^o^	Baseline	113.59 (11.97)	110.35 (10.18)	109.76 (9.54)	106.78 (7.90)	107.67 (7.88)	111.46 (8.04)	0.15
	45 days	109.39 (9.80)	105.29 (8.89)	106.90 (11.04)	105.18 (8.23)	104.68 (7.12)	107.96 (9.58)	0.51
	90 days	107.31 (11.32)	104.28 (8.46)	106.82 (8.61)	102.68 (7.68)	103.75 (8.42)	107.09 (7.52)	0.42
BCM ^p^	Baseline	31.87 (6.59)	32.29 (5.53)	33.20 (7.24)	34.94 (5.79)	33.49 (7.63)	34.68 (7.27)	0.60
	45 days	31.11 (6.31)	31.28 (5.74)	33.46 (7.16)	34.15 (5.16)	32.05 (6.84)	33.90 (6.48)	0.46
	90 days	32.09 (6.47)	31.90 (6.05)	32.54 (6.72)	35.08 (4.85)	32.26 (7.11)	34.14 (7.10)	0.61
FM ^q^	Baseline	34.21 (12.24)	31.17 (10.54)	30.85 (10.68)	28.07 (7.13)	31.04 (7.64)	33.90 (9.16)	0.31
	45 days	31.12 (11.31)	26.84 (9.10)	29.43 (10.36)	26.48 (7.76)	28.49 (6.62)	29.99 (7.85)	0.51
	90 days	29.82 (12.84)	24.82 (8.94)	27.31 (9.34)	25.46 (7.67)	26.41 (7.34)	28.71 (8.57)	0.60
FFM ^r^	Baseline	59.07 (9.99)	59.27 (9.29)	60.73 (11.47)	62.54 (8.72)	61.71 (11.76)	62.97 (12.11)	0.76
	45 days	57.62 (9.75)	57.69 (8.64)	60.71 (10.95)	61.90 (7.83)	59.71 (10.64)	61.44 (10.93)	0.61
	90 days	59.33 (10.15)	59.17 (9.27)	60.11 (10.39)	63.53 (7.60)	60.19 (11.16)	62.10 (11.63)	0.78
GLUCOSE (mmol/L)	Baseline	5.33 (0.67)	5.90 (1.33)	5.78 (1.70)	5.41 (0.75)	5.36 (0.62)	5.49 (0.67)	0.77
	45 days	5.09 (0.40)	5.77 (1.09)	5.43 (1.29)	5.15 (0.63)	5.20 (0.63)	5.35 (0.60)	0.10
	90 days	5.10 (0.40)	5.90 (1.38)	5.60 (1.53)	5.28 (1.07)	5.25 (0.89)	5.36 (0.53)	0.26
TC (mmol/L) ^s^	Baseline	5.11 (1.06)	5.11 (1.41)	5.00 (0.82)	5.31 (0.93)	5.25 (1.10)	5.34 (1.01)	0.79
	45 days	5.05 (0.85)	4.63 (1.13)	4.97 (0.82)	5.17 (0.82)	4.93 (0.88)	4.73 (1.00)	0.38
	90 days	5.04 (0.69)	4.94 (1.32)	5.32 (0.70)	5.50 (0.92)	4.95 (0.96)	4.90 (1.04)	0.30
TGL (mmol/L) ^t^	Baseline	1.19 (0.62)	1.65 (1.18)	1.39 (0.99)	1.39 (0.66)	1.63 (1.06)	1.63 (0.89)	0.78
	45 days	1.07 (0.52)	1.45 (0.93)	1.25 (0.66)	1.18 (0.74)	1.32 (1.11)	1.23 (0.84)	0.80
	90 days	1.08 (0.56)	1.42 (0.77)	1.22 (0.62)	1.38 (0.73)	1.15 (0.62)	1.30 (0.88)	0.73
HDL-C (mmol/L) ^u^	Baseline	1.22 (0.34)	1.18 (0.36)	1.13 (0.27)	1.16 (0.30)	1.14 (0.22)	1.09 (0.29)	0.80
	45 days	1.19 (0.33)	1.12 (0.36)	1.06 (0.24)	1.16 (0.30)	1.29 (0.78)	1.04 (0.32)	0.39
	90 days	1.16 (0.33)	1.24 (0.46)	1.15 (0.21)	1.24 (0.27)	1.21 (0.23)	1.11 (0.28)	0.70
HOMA-IR ^v^	Baseline	2.93 (1.69)	2.83 (1.26)	3.01 (1.68)	2.51 (1.59)	3.24 (1.76)	3.20 (1.36)	0.81
	45 days	2.14 (1.15)	2.59 (1.86)	2.05 (1.02)	1.90 (1.20)	2.05 (0.87)	1.99 (0.91)	0.60
	90 days	2.06 (1.27)	2.80 (1.46)	2.32 (1.20)	2.33 (2.33)	3.75 (6.65)	2.43 (1.38)	0.65
HBA1C (mmol/mol) ^z^	Baseline	0.06 (0.00)	0.06 (0.01)	0.06 (0.01)	0.06 (0.01)	0.06 (0.00)	0.06 (0.00)	0.83
	45 days	0.06 (0.00)	0.06 (0.01)	0.06 (0.00)	0.06 (0.01)	0.06 (0.00)	0.06 (0.01)	0.85
	90 days	0.06 (0.00)	0.05 (0.00)	0.06 (0.00)	0.06 (0.01)	0.05 (0.00)	0.06 (0.00)	0.47

^a^ CD: Control Diet based on CREA-AN guidelines; ^b^ LGIMD: Low-Glycemic-Index Mediterranean Diet; ^c^ PA1: Physical Activity 1 based on the Aerobic Activity Program; ^d^ PA2: Physical Activity 2 based on the combination of Aerobic Activity Program and Resistance Training; ^e^ PA1+LGIMD; ^f^ PA2+LGIMD; ^g^ E (kPa): Elasticity (KiloPascal); ^h^ CAP: Controlled Attenuation Parameter FibroScan^®^; ^i^ SBP: Systolic Blood Pressure; ^l^ DBP: Diastolic Blood Pressure; ^m^ BMI: Body Mass Index; ^n^ WC: Waist Circumference; ^o^ HC: Hip circumference; ^p^ BCM: Body Cell Mass; ^q^ FM: Fat Mass; ^r^ FFM: Fat-Free Mass; ^s^ TC: Total Cholesterol; ^t^ TGL: Triglycerides; ^u^ HDL-C: High-Density Lipoprotein Cholesterol; ^v^ HOMA-IR: Homeostasis model assessment for insulin resistance; ^z^ HbA1c: Glycated Hemoglobin.

**Table 3 nutrients-15-03520-t003:** Generalized Estimating Equation (GEE): Expected Values for Elasticity (kPa) by Treatment and Follow-up Time.

	β	95% CI	*p*-Value
**Follow-up**			
Baseline	0.00		
45 days	0.25	[1.02, 1.52]	0.70
3 months	1.79	[0.21, 3.38]	0.02
**Treatment**			
CD ^a^	0.00		
LGIMD ^b^	2.94	[0.67, 5.22]	0.01
PA1 ^c^	1.33	[−0.59, 3.25]	0.17
PA2 ^d^	1.21	[−0.54, 2.95]	0.17
PA1+LGIMD ^e^	1.25	[−0.51, 3.02]	0.16
PA2+LGIMD ^f^	1.14	[−0.53, 2.81]	0.18
**Treatment#time**			
LGMID # baseline	0.00		
LGMID # 45 days	−0.84	[−3.01, 1.32]	0.44
LGMID # 90 days	−2.85	[−5.24, −0.45]	0.02
PA1 # baseline	0.00		
PA1 # 45 days	0.04	[−1.89, 1.96]	0.97
PA1 # 90 days	−0.60	[−2.89, 1.70]	0.61
PA2 # baseline	0.00		
PA2 # 45 days	−0.36	[−2.06, 1.34]	0.67
PA2 # 90 days	−1.11	[−3.22, 0.99]	0.30
LGMID+PA1 # baseline	0.00		
LGMID+PA # 45 days	−1.21	[−2.93, 0.51]	0.16
LGMID+PA1 # 90 days	−2.37	[−4.39, −0.35]	0.02
LGMID+PA2 # baseline	0.00		
LGMID+PA2 # 45 days	1.02	[−0.72, 2.75]	0.25
LGMID+PA2 # 90 days	−2.21	[−4.10, −0.32]	0.02

Adjusted for gender, age, SBP: Systolic Blood Pressure, WHR: Waist-to-Hip Ratio, BMI: Body Mass Index, AA_EPA: Arachidonic Acid_Eicosapentaenoic acid ratio, GOT/GPT: Glutamic-Oxalacetic Transaminase/Glutamic-Pyruvic Transaminase ratio, ^a^ CD: Control Diet based on CREA-AN guidelines; ^b^ LGIMD: Low-Glycemic-Index Mediterranean Diet; ^c^ PA1: Physical Activity 1 based on the Aerobic Activity Program; ^d^ PA2: Physical Activity 2 based on the combination of Aerobic Activity Program and Resistance Training; ^e^ PA1+LGIMD; ^f^ PA2+LGIMD.

**Table 4 nutrients-15-03520-t004:** Marginal Mean contrasts among Intervention Arm and Follow-up Time. NUTRIATT Trial, Castellana Grotte, 2015–2016.

Follow-Up#Treatment	Contrast	95% CI
(2 vs. 1) CD ^a^	0.25	[−1.02, 1.52]
(3 vs. 2) CD	1.54	[−0.02, 3.11]
(2 vs. 1) LGMID ^b^	−0.60	[−2.43, 1.24]
(3 vs. 2) LGMID	−0.46	[−1.97, 1.05]
(2 vs. 1) PA1 ^c^	0.28	[−1.20, 1.77]
(3 vs. 2) PA1	0.91	[−0.75, 2.57]
(2 vs. 1) PA2 ^d^	−0.11	[−1.29, 1.06]
(3 vs. 2) PA2	0.79	[−0.61, 2.19]
(2 vs. 1) LGMID+PA1 ^e^	−0.96	[−2.22, 0.29]
(3 vs. 2) LGMID+PA1	0.38	[−0.75, 1.52]
(2 vs. 1) LGMID+PA2 ^f^	1.26 *	[0.03, 2.49]
(3 vs. 2) LGMID+PA2	−1.68 *	[−2.87, −0.49]

^a^ CD: Control Diet based on CREA-AN guidelines; ^b^ LGIMD: Low-Glycemic-Index Mediterranean Diet; ^c^ PA1: Physical Activity 1 based on the Aerobic Activity Program; ^d^ PA2: Physical Activity 2 based on the combination of Aerobic Activity Program and Resistance Training; ^e^ PA1+LGIMD; ^f^ PA2+LGIMD * *p*-value < 0.05.

## Data Availability

The datasets used in this study are available from the corresponding author.

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
