# Peer review of "Lifestyle Modification: Evaluation of the Effects of Physical Activity and Low-Glycemic-Index Mediterranean Diet on Fibrosis Score"

_nutrients, 2023, doi:10.3390/nu15163520_

Round 1

Reviewer 1 Report

Dear,

The manuscript “Lifestyle modification: evaluation of the effects of Physical Activity and Low Glycemic Index Mediterranean diet on fibrosis 3 score” brings interesting information about physical activity programs, Low-Glycemic Index Mediterranean Diet and Non-Alcoholic Fatty Liver Disease (NAFLD). It is very well written, clear methodology and achieves its objectives. Therefore, a minor revision of the text is necessary:

       1-     Describe in materials and methods the evaluation by bioelectrical impedance analysis;

       2- Mention figures 1 and 2 in the results and explain the reasons for the drop out shown in figure 1.

Author Response

Answers to Reviewer 1

The manuscript “Lifestyle modification: evaluation of the effects of Physical Activity and Low Glycemic Index Mediterranean diet on fibrosis 3 score” brings interesting information about physical activity programs, Low-Glycemic Index Mediterranean Diet and Non-Alcoholic Fatty Liver Disease (NAFLD). It is very well written, clear methodology and achieves its objectives. Therefore, a minor revision of the text is necessary:

  • Describe in materials and methods the evaluation by bioelectrical impedance analysis;

Thank you for the comment. We have added in the material and methods section a description of BIA. Now, it reads: Bioelectrical Impedance Analysis (BIA) was used to determine body composition indirectly by measuring the impedance, given by resistance (R) and reactance (Xc). Bioelectrical impedance was measured with a phase-sensitive touch screen impedance device (Nutrilab™, Akern, Florence, Italy), working with alternating sinusoidal electric current of 245 microampere at an operating frequency of 50 kHz (±1%).
To all subjects were asked to remain in a supine position with a leg opening of 45° compared to the median line of the body and the upper limbs positioned 30° away from the trunk. After cleansing the skin, two Ag/AgCl very low-impedance electrodes (Biatrodes, Akern Srl, Florence, Italy) were placed on the back of the right hand and two electrodes on the corresponding foot, with a distance of 5 cm between each other. To avoid disturbances in fluid distribution, subject was instructed to abstain from food and drink for >2h before the test.

2    Mention figures 1 and 2 in the results and explain the reasons for the drop out shown in figure 1.

Thank you for the comment. We have added some sentences in the statistical analysis and results sections. Now, they read:

Statistical analysis: The statistical analysis was intention-to-treat

Results: The flowchart of the study is shown in Figure 1. All drop out were due to desertion.

The results of GEE analysis are graphically displayed in Figure 2.

Reviewer 2 Report

This paper discusses various physical activity programs and their combined effect, along with a Low-Glycemic Index Mediterranean Diet, on liver fibrosis parameters in patients with NAFLD. However, a major concern regarding this paper is that the conclusions and study design appear to be too similar to the author's previous publication titled "Physical Activity and Low Glycemic Index Mediterranean Diet: Main and Modification Effects on NAFLD Score. Results from a Randomized Clinical Trial." This similarity may reduce the novelty of the current paper. From a logical standpoint, a lower NAFLD score would likely lead to a reduction in the Fibrosis score. Despite this issue, considering the significance of public health implications, I believe this paper still serves as a valuable reference for related research.

I have a few minor suggestions for improvement:

1.            The resolution of Figure 2 is too low. Kindly upload a new version with higher resolution to enhance readability.

2.            While I recall reading about the NutriAtt trial in your previous publication, I do not find a brief explanation of the trial in this paper. Please consider including a concise description for better context and understanding.

3.            I noticed that the table formats are inconsistent in your paper. It would be beneficial to ensure consistency throughout all the tables for a more professional presentation.

Thank you.

Author Response

Answers to Reviewer 2

This paper discusses various physical activity programs and their combined effect, along with a Low-Glycemic Index Mediterranean Diet, on liver fibrosis parameters in patients with NAFLD. However, a major concern regarding this paper is that the conclusions and study design appear to be too similar to the author's previous publication titled "Physical Activity and Low Glycemic Index Mediterranean Diet: Main and Modification Effects on NAFLD Score. Results from a Randomized Clinical Trial." This similarity may reduce the novelty of the current paper. From a logical standpoint, a lower NAFLD score would likely lead to a reduction in the Fibrosis score. Despite this issue, considering the significance of public health implications, I believe this paper still serves as a valuable reference for related research.

I have a few minor suggestions for improvement:

  1. The resolution of Figure 2 is too low. Kindly upload a new version with higher resolution to enhance readability.

Thank you for the comment. We have uploaded the figure with a higher resolution

  1. While I recall reading about the NutriAtt trial in your previous publication, I do not find a brief explanation of the trial in this paper. Please consider including a concise description for better context and understanding.

Thank you for the comment. We have added a concise description. Now it reads: Details about the primary study have already been published elsewhere [16]. Briefly, the NUTRIATT (NUTRItion and Ac-TiviTy) study aimed to estimate the effect of diet, PA, and their combination on NAFLD in subjects aged 30-60 years old. To do this, we enrolled patients referred to our IRCCS by the local General Practitioners. It was conducted from March 2015 to December 2016 at the National Institute of Gastroenterology, IRCCS “S. de Bellis" Research Hospital, Castellana Grotte (BA), Italy. Once enrolled and randomized, the participants were followed up by trained Nutritionist and Motor Science graduates throughout the complete follow-up. The trial, registered at clinicaltrials.gov (CT0234747696) was conducted in accordance with the Declaration of Helsinki and was approved by the Ethics Committee (Prot. No. 10/CE/De Bellis, Feb. 3, 2015).

  1. I noticed that the table formats are inconsistent in your paper. It would be beneficial to ensure consistency throughout all the tables for a more professional presentation.

Thank you for the comment. We have now reformated the tables